# Investigation and public health response to a COVID-19 outbreak in a rural resort community—Blaine County, Idaho, 2020

Eileen M. Dunne[1,2]*, Tanis Maxwell[3], Christina Dawson-Skuza[3], Matthew Burns[1], Christopher Ball[1], Kathryn Turner[1], Christine G. Hahn[1], Melody Bowyer[3], Kris K. Carter[1,4], Logan Hudson[3]

1 Division of Public Health, Idaho Department of Health and Welfare, Boise, Idaho, United States of America, 2 Epidemic Intelligence Service, Centers for Disease Control and Prevention, Atlanta, Georgia, United States of America, 3 South Central Public Health District, Twin Falls, Idaho, United States of America, 4 Center for Preparedness and Response, Centers for Disease Control and Prevention, Atlanta, Georgia, United States of America

* pgz6@cdc.gov

**Data Availability Statement:** De-identified patient-level data cannot be shared publicly because of legal restrictions. These data were collected by the

## Abstract

Blaine County, Idaho, a rural area with a renowned resort, experienced a COVID-19 outbreak early in the pandemic. We undertook an epidemiologic investigation to describe the outbreak and guide public health action. Confirmed cases of COVID-19 were identified from reports of SARS-CoV-2-positive laboratory test results to South Central Public Health District. Information on symptoms, hospitalization, recent travel, healthcare worker status, and close contacts was obtained by medical record review and patient interviews. Viral sequence analysis was conducted on a subset of available specimens. During March 13–April 10, 2020, a total of 451 COVID-19 cases among Blaine County residents (1,959 cases per 100,000 population) were reported, with earliest illness onset March 1. The median patient age was 51 years (interquartile range [IQR]: 37–63), 52 (11.5%) were hospitalized, and 5 (1.1%) died. The median duration between specimen collection and a positive laboratory result was 9 days (IQR: 4–10). Forty-four (9.8%) patients reported recent travel and an additional 37 cases occurred in out-of-state residents. Healthcare workers comprised 56 (12.4%) cases; 33 of whom worked at the only hospital in the county, leading to a 15-day disruption of hospital services. Among 562 close contacts monitored by public health authorities, laboratory-confirmed COVID-19 or compatible symptoms were identified in 51 (9.1%). Sequencing results from 34 specimens supported epidemiologic findings indicating travel as a source of SARS-CoV-2, and identified multiple lineages among hospital workers. Community mitigation strategies included school and resort closure, stay-at-home orders, and restrictions on incoming travelers. COVID-19 outbreaks in rural communities can disrupt health services. Lack of local laboratory capacity led to long turnaround times for COVID-19 test results. Rural communities frequented by tourists face unique challenges during the COVID-19 pandemic. Implementing restrictions on incoming travelers and other mitigation strategies helped reduce COVID-19 transmission early in the pandemic.

Idaho Department of Health and Welfare as part of reportable disease surveillance under Idaho administrative code (IDAPA 16.02.10 https://adminrules.idaho.gov/rules/current/16/160210.pdf). Use of these data for other purposes requires approval from the Idaho Division of Public Health. De-identified patient data can be requested from the Idaho Division of Public Health by contacting the Bureau of Communicable Diseases Epidemiology Section at Epimail@dhw.Idaho.gov. Other types of data included in this paper are publicly available. SARS-CoV-2 sequence data have been uploaded to the GISAID database, with accession numbers provided in S1 Table. Data on the estimated proportion of Blaine County residents staying at home are available from Safegraph, Inc (https://docs.safegraph.com/docs/social-distancing-metrics). Census block group data are available from the United States Census Bureau (https://data.census.gov/cedsci/).

**Funding:** The authors received no specific funding for this work.

**Competing interests:** The authors have declared that no competing interests exist.

## Introduction

Rural residents might be particularly vulnerable to the novel coronavirus disease (COVID-19) pandemic. Approximately 15% of Americans live in rural areas, and rural populations in the United States tend to be older, have higher rates of underlying conditions, and less access to health services [1, 2]. The health gap between rural and urban US was highlighted by a 2017 report identifying higher age-adjusted death rates for the five leading causes of death in non-metropolitan areas [3]. Rural health departments face unique challenges related to serving the needs of populations that experience poorer health outcomes in large geographical areas with limited resources [2]. Survey data indicate that the COVID-19 pandemic has broad, negative impacts on the well-being of rural populations in the American West [4].

The state of Idaho, located in the northwest region of the United States, has an estimated population of approximately 1.79 million, of which 32.4% reside in rural areas [5]. Idaho is divided into 44 counties, 32 (73%) of which are classified as rural [6], and seven autonomous public health districts comprising 4 to 8 counties each. On March 13, 2020, a case of confirmed COVID-19 in a Blaine County resident was reported to South Central Public Health District (SCPHD); this was the second case identified in Idaho. Blaine County is a scenic, mountainous rural area with an estimated 23,021 residents, of whom 23.5% are Hispanic or Latino [7]. It is a popular weekend and vacation destination, with abundant outdoors and cultural activities, including the Sun Valley Resort, an internationally renowned ski area. An estimated 32% of housing units are occupied seasonally as second homes or short-term rentals [8]. The county is economically diverse, with a 2018 median household income of $51,968 and a poverty rate of 14% [9]. In this report, we present the epidemiology of the COVID-19 outbreak in Blaine County including patient characteristics, hospitalization rates, travel history, and healthcare worker status. Sequence analysis of available SARS-CoV-2 specimens supported epidemiological findings. We examine positive COVID-19 test turnaround times and describe the interruption of hospital services, and the public health response.

## Methods

### Case identification and investigation

A confirmed COVID-19 case was defined as detection of SARS-CoV-2 RNA in a clinical specimen (nasopharyngeal swab, nasal swab, or oropharyngeal swab) by using real-time reverse transcriptase polymerase chain reaction (RT-PCR). Confirmed COVID-19 cases were identified via reporting of SARS-CoV-2-positive laboratory tests to SCPHD. Case information, including electronic laboratory reports, was collected and stored using Idaho's National Electronic Disease Surveillance System (NEDSS) Base System (NBS), a Centers for Disease Control and Prevention (CDC)-supported integrated disease surveillance system. Initial investigation determined the usual residence of the patient to confirm reporting jurisdiction, as cases in out-of-state residents are not included in official case counts per standard reporting procedures [10].

Data on symptom onset, hospitalization, recent travel, healthcare worker status, and close contacts were obtained by medical record review and interviews with patients or their proxies. Healthcare workers were defined as paid employees or volunteers who worked at a hospital or other healthcare facility, pharmacists, emergency medical service responders, and firefighters with emergency medical technician certification.

Initially, close contacts were defined as household members and others who spent ten minutes or more within a six feet of a patient from 1 day prior to symptom onset in the patient with COVID-19; this time frame was expanded to 2 days prior to symptom onset following

further evidence of presymptomatic transmission [11]. Close contacts were monitored through 14 days after exposure.

### Viral sequence analysis

Following diagnostic testing at the Idaho Bureau of Laboratories, aliquots of viral transport media were preserved at -80˚C until a subset was selected for sequencing. Selected samples were preserved in Zymo DNA/RNA Shield, a proprietary reagent which stabilizes nucleic acid samples at ambient temperatures, and sent to the University of New Mexico Center for Global Health for whole genome sequence analysis. Sequencing was conducted using the ARCTIC protocol, version 3 and assembly was performed by the Center for Global Health [12]. All finished sequences were uploaded with required metadata to the GISAID EpiCoV™ (S1 Table).

All complete, high coverage, SARS-CoV-2 sequences from samples collected during March 1–March 12, 2020 in the USA were downloaded from the GISAID database (S2 Table) [13]. This period represents the 10 days prior to the first laboratory confirmed COVID-19 case in Idaho. Idaho sequences from Blaine County were BLASTed against this subset of sequences using the MEGABLAST tool within Bionumerics. The top two hits by sequence identity were selected for further analysis. The BLAST hits, Blaine County sequences, and sequences from other Idaho counties (collected March 1–April 1, 2020) were compared using Bionumerics Multiple Sequence Alignment Tool and clustered using a Minimum Spanning Tree.

### Community mitigation measures

Information on school and business closures and governmental orders was obtained from press releases. Data on the estimated proportion of county residents staying at home were made publicly available by SafeGraph, Inc [14].

### Data analysis

Data were analyzed using Stata version 14.2 and graphs created in Excel. Data were reported as percentages for categorical variables and median, interquartile range (IQR), and range for continuous variables. Multivariable logistic regression was used to evaluate variables associated with hospitalization, with robust standard error to account for possible lack of independence, and results reported as adjusted odds ratios (aOR) with 95% confidence intervals (95% CI). Methods for census block group analysis are described in S3 Table.

### Research determination

COVID-19 is a reportable disease under Idaho Department of Health and Welfare Rules, IDAPA 16.02.10. Case investigation, data collection, and analysis were conducted for public health purposes. Data were fully anonymized prior to analysis. This project was reviewed by the Center for Surveillance, Epidemiology, and Laboratory Services Human Subjects Contact at the CDC. The project was determined to meet the requirements of public health surveillance covered by the U.S. Department of Health and Human Services Policy for the Protection of Human Research Subjects as defined in 45 CFR 46.102 (https://www.hhs.gov/ohrp/regulations-and-policy/regulations/45-cfr-46/index.html), and the decision was made that this project was nonresearch and did not require ethical review by the CDC Human Research Protection Office. Ethical approval was waived and informed consent was not required.

## Results

Four weeks after the first case was identified, a total of 452 confirmed cases were reported among Blaine County residents. One case was excluded because the patient was temporarily residing in another state during the outbreak. An additional 37 cases were identified among out-of-state residents who were tested in Blaine County; these included 26 residents of areas where COVID-19 outbreaks were known to have been occurring (15 patients from King County, Washington and 11 patients from counties in California where community spread had occurred).

Blaine County experienced one of the highest rates of COVID-19 cases per capita (1,959/100,000) in the US at the time of this investigation. Of the 451 patients, 239 (53.0%) were female. The median age at onset was 51 years (IQR: 38–63 years), with 5 (1.1%) patients aged <18 years, 169 (37.4%) aged 18–44 years, 172 (38.1%) aged 45–64 years, and 106 (23.5%) aged ≥65 years. Race and ethnicity data indicated that 332 (73.5%) patients were white (non-Hispanic or Latino), 9 (2.0%) other race, and 73 (16.2%) Hispanic or Latino, with data missing for 37 (8.2%) patients.

Among 447 patients with symptom data available, 446 (99.8%) reported at least one symptom. Most commonly reported symptoms included cough (n = 299, 66.9%), fever (measured or subjective; n = 275, 61.5%), myalgia or body aches (n = 208, 46.5%), fatigue (n = 205, 45.9%), headache (n = 164, 36.7%), and shortness of breath (n = 137, 30.6%). Other reported symptoms included chills (n = 111, 24.8%), loss of taste or smell (n = 101, 22.6%), sore throat (n = 83, 18.6%), diarrhea (n = 83, 18.6%), congestion (n = 81, 18.1%), pain or tightness in chest (n = 55, 12.4%), nausea or vomiting (n = 54, 12.1%), loss of appetite (n = 39, 8.7%), and runny nose (n = 36, 8.1%). The median total number of symptoms reported per patient was 4 (IQR: 3–6, range = 0–11); 187 (41.8%) of patients reported having five or more symptoms.

A total of 52 (11.5%) patients were hospitalized. The number and proportion of patients who were hospitalized increased with age, ranging from no (0%) patients aged <18 years, 5 (3.0%) patients aged 18–44 years, 14 (8.1%) aged 45–64 years, and 33 (31.4%) persons aged ≥65 years. Although information on underlying medical conditions was not available, multivariable regression analysis to examine factors associated with hospitalization found that age group 45–64 years (aOR 4.0, 95%CI 1.0–15.5), age group ≥65 years (aOR 23.0,95%CI 5.6–94.4), Hispanic or Latino ethnicity (aOR 3.6, 95%CI 1.2–10.5), and having five or more symptoms (aOR 3.1,1.5–6.4) were associated with increased odds of hospitalization (S4 Table).

The median length of admission was 5 days (IQR: 2–11 days, range = 1–38 days). Of 50 hospitalized patients with available data, 21 (42%) were admitted to the intensive care unit. Overall, five (1.1%) patients died; all were aged >60 years and 4 (80%) were male.

The median duration between symptom onset and specimen collection was 5 days (IQR: 2–8; range = 0–24). The median duration from specimen collection to reporting of a positive SARS-CoV-2 laboratory result was 9 days (IQR: 4–10 days; range = 1–22 days): 2 days (IQR: 1–3; range = 1–5) for high-priority specimens (e.g., from healthcare workers and hospitalized patients) tested by the Idaho Bureau of Laboratories (n = 41), and 9 days (IQR: 5–10; range = 1–22) for specimens tested at commercial laboratories (n = 410). In total, the median time between illness onset and reporting of a SARS-CoV-2 positive test was 13 days (IQR: 10–16; range = 2–34). Idaho Bureau of Laboratories began SARS-CoV-2 testing on March 2, and regional commercial laboratories began SARS-CoV-2 testing on March 5. Testing availability increased in Blaine County following the March 17 opening of a COVID-19 screening and testing center adjacent to the sole hospital in the county (Hospital A) and operated by their health system.

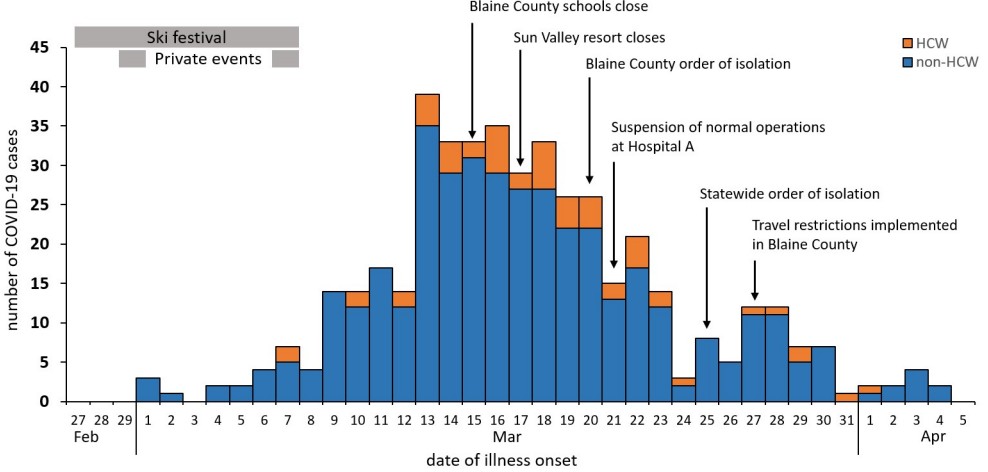

**Fig 1. Epidemic curve showing cases of COVID-19 in Blaine County, Idaho residents by date of illness onset (n = 447).** Cases in health care workers (HCW) are shown in orange and non-HCW in blue. Onset dates from cases reported from March 13–April 10, 2020 are included on the graph. Dates of events linked to 11 cases are indicated with gray boxes, and implementation dates for community mitigation measures are shown with arrows.

At least 11 cases occurred in Blaine County residents who attended three events (a ski festival and two private events) held during February 27 to March 7 that attracted many out-of-state and international travelers. Data on recent travel history and healthcare worker status were available for 450/451 (99.8%) cases. Of these, 44 (9.8%) patients reported travel from another state or country during the 2 weeks prior to symptom onset. In total, 56 (12.4%) cases were among healthcare workers, 33 of whom worked at Hospital A, a critical access hospital (a designation given to eligible rural hospitals by the Centers for Medicare and Medicaid Services). Affected hospital staff held clinical and non-clinical roles throughout the hospital.

Fig 1 illustrates the epidemiologic curve for COVID-19 illness onset among 447 patients with available data, including healthcare workers, and shows dates of the ski festival and two private events and community mitigation measures. The epidemic curve indicated a peak in illness onset in mid-March, consistent with exposures occurring during the two weeks prior. Hospital A suspended normal operations on March 20, one day after community spread in Blaine County was announced. Hospital A's emergency department and COVID-19 screening and testing center remained open; however, non-emergent appointments and procedures were postponed, inpatient services discontinued, and affiliated community clinics closed. Patients requiring admission were transported by emergency medical services to the nearest regional hospital located 78 miles away. Hospital A resumed limited services on April 3. S1 Fig depicts hospitalizations over time by date of admission.

Analysis of SARS-CoV-2 sequences was conducted on a convenience sample of available specimens from Idaho patients diagnosed with COVID-19, including 23 from Blaine County residents and 11 from other counties in Idaho. Several genomes from Blaine County residents were closely related to sequences identified in other states including New York, Louisiana, Mississippi, and Rhode Island (Fig 2). Three genomes from patients who were residents of other Idaho counties but spent time in Blaine County prior to illness onset were closely related to genomes from Blaine County residents (Fig 2). Analysis of viral sequences from Hospital A staff members identified two SARS-CoV-2 lineages (B.1 and A.1), suggesting that the outbreak among Hospital A staff was not due to a single source.

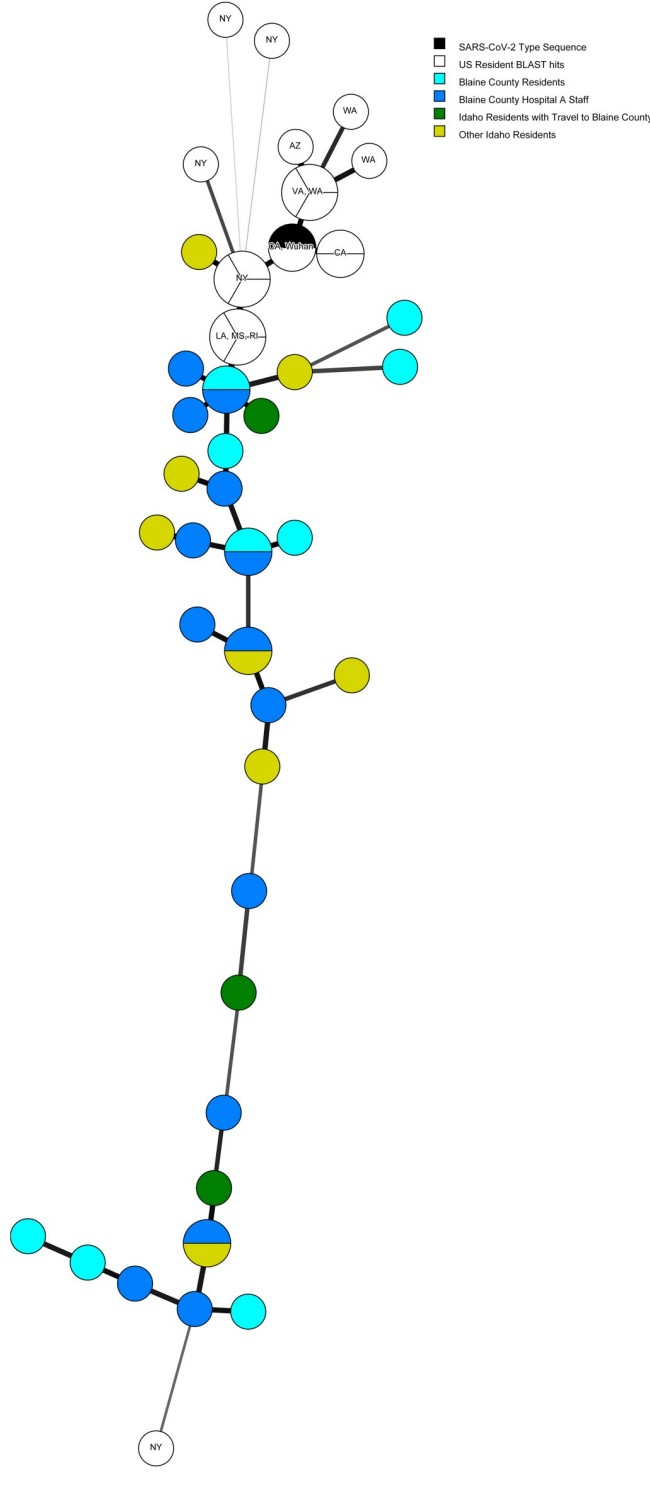

**Fig 2. Minimum spanning tree showing SARS-CoV-2 sequences from Blaine County residents who worked at Hospital A (n = 14, dark blue), Blaine County residents who did not work at Hospital A (n = 9, light blue), residents of other Idaho counties with no travel to Blaine County (n = 7, light green), residents of other Idaho counties who traveled in Blaine County prior to illness (n = 3, dark green), related sequences from other US states identified by BLAST (n = 19, white), and the reference sequence from Wuhan, China (black).** GSAID accession numbers are listed in S1 Table.

Census block group estimates were used to examine geographic distribution and community characteristics for 402 (89.1%) COVID-19 patients whose addresses were matched to a census block group, which can serve as a proxy for neighborhood. At least one case occurred in each of the 13 census block groups in Blaine County (S3 Table). Community characteristics varied substantially by census block group. Over half (n = 216, 53.7%) of cases occurred among residents of four census block groups. Within these four census block groups, the proportion of the population that are Hispanic or Latino ethnicity ranged from 4.6% to 47.1% and the proportion of persons in renter-occupied housing ranged from 11.1% to 49.2%.

Public health authorities, the county government, and local stakeholders undertook several measures to contain the outbreak and limit community transmission. SCPHD conducted contact tracing and monitored 562 close contacts of Blaine County cases; 22 (3.9%) of these close contacts tested positive for SARS-CoV-2 and an additional 29 (5.2%) became symptomatic but were not tested. During March 9–April 10, SCDPH staff worked 4,074 hours on COVID-19 emergency response. This response supported all eight counties in the jurisdiction, however 452/495 (91.3%) cases occurred in Blaine County. Volunteers spent 158 hours conducting contact monitoring activities. Blaine County School District closed school buildings on March 14. Sun Valley Resort closed for the season on March 16. The Idaho Department of Health and Welfare (IDHW) issued an order of isolation for Blaine County residents on March 20. Central District Health, whose jurisdiction borders SCPHD and includes the state capital of Boise (less than three hours driving time from Sun Valley Resort) issued a statement on March 22 instructing anyone with recent travel to Blaine County to shelter in place following identification of four COVID-19 cases among residents who had recently been in Blaine County. On March 25, IDHW issued a statewide order to self-isolate. On March 27, Blaine County issued an order with additional restrictions including the prohibition of non-essential travel into or out of Blaine County and 14-day self-quarantine for residents or visitors coming from out of state. S2 Fig shows the timing of these measures along with cumulative case counts. Publicly accessible data from SafeGraph, Inc. reported county-level percentage of people staying home all day (based on GPS data from anonymous mobile devices) indicating that the proportion of people staying at home increased steadily from March 15 through early April (Fig 3).

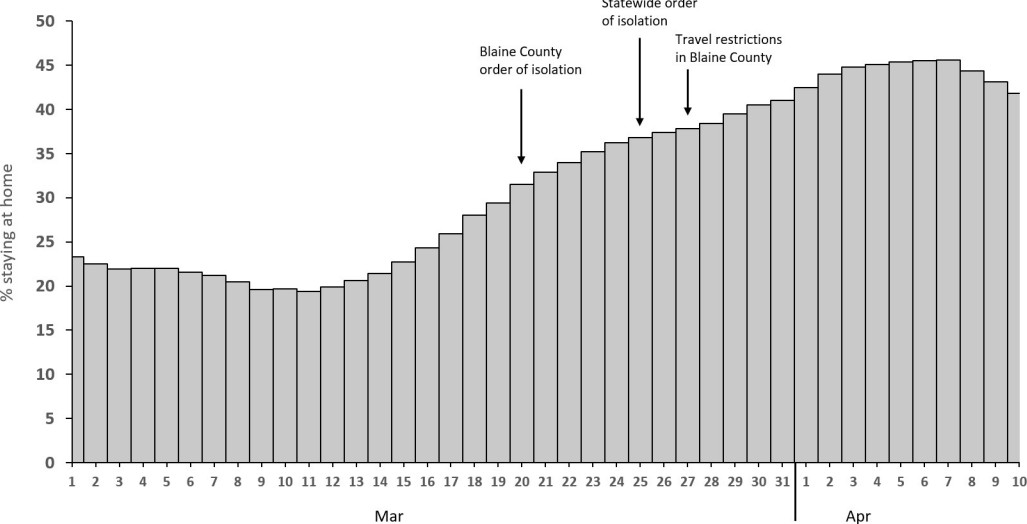

**Fig 3. Daily estimates of the percent of people in Blaine County, Idaho staying at home all day during March 1–April 10, 2020.** Data were obtained from SafeGraph, Inc. and are based upon global positioning system data from mobile devices. Dates of isolation orders and travel restrictions are indicated on the graph.

## Discussion

The start of the outbreak in Blaine County was likely linked to travel to the ski resort, high rates of seasonal residence, and three events held during late February and early March 2020 that attracted numerous out-of-state attendees. COVID-19 outbreaks have been identified in mountain resort communities elsewhere in the US and in Europe [15, 16]. Several published reports demonstrate how travel and events contributed to COVID-19 outbreaks. In the district of Tirschenreuth, Germany, travel to ski resorts and attendance at traditional beer festivities were implicated as early drivers of transmission for a COVID-19 outbreak that began in mid-March 2020, which was controlled following implementation of a 14-day quarantine for returning travelers and cancellation of large events [17]. In Maine, a 55-person wedding in a rural town with several out-of-state attendees was linked to 177 COVID-19 cases as well as outbreaks at a long-term care facility and a correctional facility [18]. Several COVID-19 cases identified in Blaine County occurred in residents of King County, Washington and counties in California where community spread had occurred prior to detection of the first COVID-19 case in Idaho. Cases in out-of-state residents, although not included in official case counts, are indicative of frequent travel from the west coast and likely contributed to the level of transmission observed. Blaine County was the first county in Idaho to announce community spread of COVID-19 despite being the 17th most populous county. Second home ownership, common in resort communities such as Blaine County, has been linked to the spread of COVID-19 from urban to rural areas in several countries including the United States [19, 20]. Improving strategies to detect and monitor SARS-CoV-2 infections among temporary residents might help resort communities better understand their role in transmission and prepare for potential healthcare and public health impacts.

Interstate spread has been implicated in SARS-CoV-2 importations in other regions of the United States. Sequence analysis of SARS-CoV-2 found that 7 of 9 viral genomes from early COVID-19 cases in Connecticut clustered with a clade dominated by viruses from cases in Washington state [21]. Sequence analysis of SARS-CoV-2 from Blaine County supported the epidemiological findings, as several genomes clustered with viruses from other states, suggestive of links to travel. Because certain states were likely overrepresented in the database of available sequences from early in the pandemic, and Idaho sequences indicate multiple introductions of SARS-CoV-2, the geographic source of the outbreak could not be identified. Limitations on the use of viral sequencing for this epidemiologic investigation include that only a subset of specimens tested and stored at the state public health laboratory were available for sequence analysis, as commercial laboratories typically do not store specimens.

Rural counties have fewer healthcare workers and facilities compared with urban areas, making rural areas vulnerable to staff shortages and facility closures that could reduce access to health care. The high number of affected workers from Hospital A led to temporary cessation of inpatient services at the only hospital in Blaine County. Sequencing analysis of SARS-CoV-2 from Hospital A staff indicated multiple exposures rather than a point-source outbreak caused by a single viral strain, and it was not possible to determine whether staff were infected in the community or in the workplace. Work exclusion of Hospital A staff who were identified as close contacts of COVID-19 cases also contributed to the staffing shortage. As Hospital A is part of a larger and well-resourced health system, it was able to remain partially open and received support from partner hospitals. Hospital planning for community spread of COVID-19, vaccination of healthcare workers, and developing strategies for mitigating staffing shortages are critical for maintaining healthcare access in rural areas. In Thailand, a rural hospital closed following confirmed cases of COVID-19 in three medical personnel and quarantine of the remaining 21 medical personnel of the hospital [22]. A survey of COVID-19 preparedness

among hospitals in Idaho identified inconsistent implementation of CDC guidelines for infection prevention, and only 8 of 21 (38%) critical access hospitals had a written respiratory protection program [23]. Limited isolation facilities pose a challenge, as only 2 (10%) critical access hospitals had an airborne infection isolation room.

The CDC has developed relevant resources including hospital preparedness checklist for COVID-19 (https://www.cdc.gov/coronavirus/2019-ncov/hcp/hcp-hospital-checklist.html). Telehealth is another strategy that can be leveraged to help rural hospitals during the COVID-19 pandemic by expanding healthcare capacity, reducing potential exposures of healthcare workers, and linking with tertiary hospitals to provide additional staffing support and reduce unnecessary patient transfers [24].

Case identification and contact tracing efforts were hindered by long lag times between symptom onset and receipt of laboratory results. Many contacts were not notified of their potential exposure until the second week of the 14-day isolation period. Delays in testing turn-around times might be more prominent in predominantly rural states like Idaho, where commercial testing during the investigation time frame was only available at regional laboratories located in other states. Public health laboratory support enabled faster turnaround times for high-priority specimens. Expansion of diagnostic testing availability, including at hospital and commercial diagnostic laboratories and point-of-care rapid tests, has subsequently led to shorter turnaround times in Idaho. A commentary advocated for expanded testing in rural areas, as the authors' analysis found that states with higher prevalence of COVID-19 risk factors (hypertension, diabetes, and lung cancer), which tend to be more common in rural populations, had lower overall testing rates, and medically vulnerable people in rural areas have greater potential for severe outcomes [25]. Contact tracing, a key component of the public health response to COVID-19, depends on the ability to detect COVID-19 cases in a timely manner.

Part-time residents are typically not included in population estimates, limiting the accuracy of disease incidence estimates in areas where tourism and seasonal residence are common, such as Blaine County, where the ski season attracts tourists and seasonal workers. Cases of COVID-19 among visitors to Blaine County who were tested outside of Idaho were not captured in our analysis, therefore we were not able to assess how exposures at the ski resort and elsewhere in Blaine County might have contributed to the spread of COVID-19 in other states. Additional limitations include that testing during the time of this investigation was primarily conducted on symptomatic persons, and testing of children was relatively uncommon. The rapid rise in reported cases seen in late March and early April might reflect expanded testing availability as well as disease transmission. Data on underlying conditions were not available, and relationships between patients (other than cases in contacts under monitoring) were not systematically captured, although several household clusters were identified. Geocoding patient addresses and census block group data indicated that the outbreak was not limited to a specific geographical region of the county, and it affected communities of varying socio-economic characteristics. Of patients with available race and ethnicity information, 17.6% were Hispanic or Latino, compared with 23.5% of the overall Blaine County population. This slight underrepresentation might be because of several factors including disparities in access to health care and testing [26]. However, Hispanic or Latino ethnicity was associated with increased odds of hospitalization, consistent with reports highlighting the disproportionate impact of COVID-19 on the Hispanic or Latino population in the United States [27, 28].

Evidence from four US metropolitan areas indicated that community mobility declined following implementation of stay-at-home orders [29]. Consistent with these findings, the estimated percentage of people staying at home in Blaine County, obtained via anonymous cellular phone data, increased following orders of isolation. In our experience, after the

implementation of community mitigation measures, most contacts identified were household members. The Blaine County outbreak attracted intense local and national media coverage that could have encouraged residents' adherence to stay-at-home orders [30, 31]. It is challenging to directly assess the impact of community mitigation measures on disease transmission, because of the time lag between exposure, symptom onset, and case detection, and the multitude of unmeasured factors that might influence COVID-19 transmission and case detection. In Blaine County, mitigation strategies including closure of the ski resort and a county-wide isolation order began within a week of the first reported case. The increase in cumulative cases in Blaine County began to slow approximately two weeks following implementation of the county order of isolation. Social distancing and stay-at-home orders were valuable tools for reducing transmission early in the pandemic before COVID-19 treatments or vaccines were available.

## Conclusions

Rural communities frequented by travelers and seasonal residents, like Blaine County, can be heavily impacted by COVID-19, leading to disruption in available health services. Rural hospitals should develop pandemic preparedness plans that include strategies for mitigating staffing shortages. Expanding COVID-19 testing availability and speed will help rural health departments and healthcare systems detect and respond to COVID-19 outbreaks. Closing tourist attractions and implementing restrictions on incoming travelers in addition to stay-at-home orders and other policies targeting local residents were effective strategies for limiting community spread in rural areas prior to widespread availability of COVID-19 vaccines.

## Supporting information

**S1 Table. SARS-CoV-2 sequences from Idaho and others included in Fig 2.**
(PDF)

**S2 Table. SARS-CoV-2 sequences from GSAID used in analyses.**
(PDF)

**S3 Table. Distribution of confirmed COVID-19 cases (n = 402) and census block group characteristics for Blaine County, Idaho.**
(PDF)

**S4 Table. Univariable and multivariable analysis of patient characteristics associated with hospitalization.**
(PDF)

**S1 Fig. Hospitalization of Blaine County, Idaho residents for COVID-19 by date of admission (n = 52).**
(PDF)

**S2 Fig. Timeline of the COVID-9 outbreak in Blaine County showing cumulative COVID-19 case counts and implementation dates for community mitigation measures.**
(PDF)

## Acknowledgments

We acknowledge staff and volunteersat South Central Public Health District, and the Idaho Department of Health and Welfare, Division of Public Health, Bureau of Communicable Disease Prevention and the Idaho Bureau of Laboratories. We thank Mubarak Tukur, Sheri

Tolley, and Donald Rock and the Infection Prevention team at St. Luke's Health System. We acknowledge Daryl Doman and Darrell Dinwiddie from the University of New Mexico for viral sequencing. We thank Melinda Bauman, Dan Schaffer, Mayra Vasquez-Alvarez, and Jared Bartschi for assistance with data collection, and Bozena Morawski, Chris Johnson, and Kris Bisgard for reviewing the manuscript.

## Author Contributions

**Conceptualization:** Christine G. Hahn, Kris K. Carter, Logan Hudson.

**Data curation:** Matthew Burns, Kathryn Turner.

**Formal analysis:** Eileen M. Dunne, Matthew Burns.

**Investigation:** Eileen M. Dunne, Tanis Maxwell, Christina Dawson-Skuza, Logan Hudson.

**Methodology:** Christopher Ball, Kris K. Carter.

**Supervision:** Kathryn Turner, Christine G. Hahn, Melody Bowyer, Kris K. Carter, Logan Hudson.

**Writing – original draft:** Eileen M. Dunne.

**Writing – review & editing:** Tanis Maxwell, Christina Dawson-Skuza, Matthew Burns, Christopher Ball, Kathryn Turner, Christine G. Hahn, Melody Bowyer, Kris K. Carter, Logan Hudson.

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
