## [Decision Letter · Decision Letter 0]

1 Mar 2021

PONE-D-21-03832

Investigation and public health response to a COVID-19 outbreak in a rural resort community — Blaine County, Idaho, 2020

PLOS ONE

Dear Dr. Dunne,

Thank you for submitting your manuscript to PLOS ONE. After careful consideration, we feel that it has merit but does not fully meet PLOS ONE’s publication criteria as it currently stands. Therefore, we invite you to submit a revised version of the manuscript that addresses the points raised during the review process.

We look forward to receiving your revised manuscript.

Kind regards,

Shinya Tsuzuki, MD, MSc

Academic Editor

PLOS ONE

Journal Requirements:

2. Please confirm that your IRB board waived the need for ethical approval. Moreover, in ethics statement in the manuscript and in the online submission form, please provide additional information about the patient records used in your retrospective study. Specifically, please ensure that you have discussed whether all data were fully anonymized before you accessed them and/or whether the IRB or ethics committee waived the requirement for informed consent.

3.We note that you have indicated that data from this study are available upon request. PLOS only allows data to be available upon request if there are legal or ethical restrictions on sharing data publicly. For information on unacceptable data access restrictions, please see http://journals.plos.org/plosone/s/data-availability#loc-unacceptable-data-access-restrictions.

Additional Editor Comments:

Both reviewers gave insightful suggestions then please revise your manuscript accordingly.

Reviewers' comments:

Reviewer's Responses to Questions

**Comments to the Author**

1. Is the manuscript technically sound, and do the data support the conclusions?

Reviewer #1: Yes

Reviewer #2: Yes

2. Has the statistical analysis been performed appropriately and rigorously? 

Reviewer #1: No

Reviewer #2: Yes

3. Have the authors made all data underlying the findings in their manuscript fully available?

Reviewer #1: No

Reviewer #2: Yes

4. Is the manuscript presented in an intelligible fashion and written in standard English?

Reviewer #1: Yes

Reviewer #2: Yes

5. Review Comments to the Author

Reviewer #1: Summary

This report summarizes an outbreak investigation of COVID-19 in a rural county in Idaho with a popular resort community early in the course of pandemic-related events in the US. This report highlights some of the challenges faced early in the pandemic response, when testing was limited and public health action delayed due to the lack of widespread availability of rapid tests. This report is well written, and represents a thorough investigation, including case interview/medical record review, contact tracing, sequencing, etc. This report does not add much insight into COVID-19 epidemiology; however, it does provide insight into the issues challenging the response early in the pandemic and in rural areas.

Major Issues

This report is very well written and covers many important aspects of a COVID-19 outbreak investigation despite the limitations of being early in the pandemic when less was known about risk factors, asymptomatic spread, etc. I have very few suggestions to improve it, other than the few provided below.

The report uses only descriptive statistics to describe the outbreak. However, there is a place for analytic epidemiology in this report. The authors should consider conducting multivariable logistic regression to examine risk factors for hospitalization and report ORs with 95% CIs instead of just frequency of age ranges for those hospitalized.

Minor Issues

Suggest reporting frequency/percent of individuals reporting one and multiple symptoms (e.g., 1, 2-4, 5+) or something similar. Were there any set of symptoms that most commonly occurred together?

Reviewer #2: The authors have reported on an investigation of COVID-19 in a rural resort community. The report is comprehensive of many aspects of the outbreak and response. However, it lacks the focus that would make it a useful scientific paper. For example, it is noted that the outbreak occurred after a ski event at the local ski resort. However, there is no real assessment of how that event contributed to spread across the county. The rural nature of the county is emphasized, as is the resort nature of the community. These raise separate, important issues with regard to transmission risk and control, and neither is fully addressed. The abstract describes the occurrence of 452 cases during March 13-April 10. However, the epicurve (figure 1) identifies more than 60-70 cases with onset of symptoms before March 13. The epi-curve and the descriptions of the outbreak are difficult to reconcile. I think that it would be useful to focus on the impact of the ski resort in the initiation of the outbreak. It would also be useful to highlight the interactions of the control measures on the epi-curve (fig. 1). If the epi-curve could be labeled with all of the key time line events, (ski festival, school closure, resort closure, county isolation order, state isolation order, travel restriction) it would provide a clear visual guide to the progression of the outbreak. This would be a better way to relate control measures to case incidence than is achieved in figure 3. The use of cumulative frequency data make it difficult to track the rate of new case onsets. It would be an even bigger bonus if the cases identified by sequence cluster could also be indicated on the epi-curve. I recognize that may not be feasible.

It seems that approximately 1/3 of the population of the county at any given time may be made up by visitors from out of state. It would be useful to try to estimate the rate of illness in this ephemeral group, or at least try to better estimate the impact of visitors on transmission dynamics. It is critical for similar resort communities to try to plan for potential disease introductions from tourists, and to have plans for surveillance and control of spread related to these temporary residents. It may be true that such cases are not typically counted as cases in the temporary community, but that does not diminish the importance of being able to track and account for them.

There are descriptions of definitions of close contacts on p.4, and changes in the definition. There does not seem to be any use of this data in the paper. If it is not being used for analytical purposes, is it needed in the methods? Similarly, there is substantial description of census blocks in the methods. The results do not present more than a cursory analysis of the census block data, that could be omitted. How do the census blocks relate to housing for guests of the resort?

The introductions and discussions are very broad. Given the vast literature that is being published on the pandemic, a narrower focus on the impact of the ski resort on the community would enhance the interest and usefulness of this paper.

6. PLOS authors have the option to publish the peer review history of their article (what does this mean?). If published, this will include your full peer review and any attached files.

Reviewer #1: No

Reviewer #2: No

---

## [Author Response · Author response to Decision Letter 0]

17 Mar 2021

Authors’ response: We have formatted our manuscript in accordance with PLOS ONE’s style requirements. 

2. Please confirm that your IRB board waived the need for ethical approval. Moreover, in ethics statement in the manuscript and in the online submission form, please provide additional information about the patient records used in your retrospective study. Specifically, please ensure that you have discussed whether all data were fully anonymized before you accessed them and/or whether the IRB or ethics committee waived the requirement for informed consent.

Authors’ response: We have provided additional information on research determination including the waiver of ethical approval and informed consent in the online submission form and in the ethics section of the manuscript. Please see the revised text below:

COVID-19 is a reportable disease under Idaho Department of Health and Welfare Rules, IDAPA 16.02.10. Case investigation, data collection, and analysis were conducted for public health purposes. Data were fully anonymized prior to analysis. This project was reviewed by the Center for Surveillance, Epidemiology, and Laboratory Services Human Subjects Contact at the CDC. The project was determined to meet the requirements of public health surveillance covered by the U.S. Department of Health and Human Services Policy for the Protection of Human Research Subjects as defined in 45 CFR 46.102 (https://www.hhs.gov/ohrp/regulations-and-policy/regulations/45-cfr-46/index.html), and the decision was made that this project was nonresearch and did not require ethical review by the CDC Human Research Protection Office. Ethical approval was waived and informed consent was not required.

3.We note that you have indicated that data from this study are available upon request. PLOS only allows data to be available upon request if there are legal or ethical restrictions on sharing data publicly. For information on unacceptable data access restrictions, please see http://journals.plos.org/plosone/s/data-availability#loc-unacceptable-data-access-restrictions.

Authors’ response: Information on ethical and legal restrictions to sharing de-identified data has been provided in the revised cover letter as instructed.

Additional Editor Comments:

Both reviewers gave insightful suggestions then please revise your manuscript accordingly.

Authors’ response: The manuscript has been revised in response to reviewer suggestions, with point-by-point responses to reviewer comments provided below. We thank the reviewers for their prompt and thorough review and appreciate their comments and feedback. 

Reviewers' comments:

Reviewer #1: Summary

This report summarizes an outbreak investigation of COVID-19 in a rural county in Idaho with a popular resort community early in the course of pandemic-related events in the US. This report highlights some of the challenges faced early in the pandemic response, when testing was limited and public health action delayed due to the lack of widespread availability of rapid tests. This report is well written, and represents a thorough investigation, including case interview/medical record review, contact tracing, sequencing, etc. This report does not add much insight into COVID-19 epidemiology; however, it does provide insight into the issues challenging the response early in the pandemic and in rural areas.

Major Issues

This report is very well written and covers many important aspects of a COVID-19 outbreak investigation despite the limitations of being early in the pandemic when less was known about risk factors, asymptomatic spread, etc. I have very few suggestions to improve it, other than the few provided below.

The report uses only descriptive statistics to describe the outbreak. However, there is a place for analytic epidemiology in this report. The authors should consider conducting multivariable logistic regression to examine risk factors for hospitalization and report ORs with 95% CIs instead of just frequency of age ranges for those hospitalized.

Authors’ response: Unfortunately, we did not collect data on underlying medical conditions, many of which have been associated with increased risk of severe illness from COVID-19 including hospitalization. Therefore, we were not able to conduct a thorough assessment of risk factors for hospitalization. However, we did conduct a multivariable analysis as suggested including age group, sex, race/ethnicity, healthcare worker status, and number of symptoms, with results summarized in the text (lines 156-160) and shown in S4 Table.

Minor Issues

Suggest reporting frequency/percent of individuals reporting one and multiple symptoms (e.g., 1, 2-4, 5+) or something similar. Were there any set of symptoms that most commonly occurred together?

Authors’ response: We have added data on the number of symptoms reported to lines 151-152. Aside from the fact that commonly reported symptoms were often reported together, we did not observe any particular patterns regarding symptoms.

Reviewer #2: The authors have reported on an investigation of COVID-19 in a rural resort community. The report is comprehensive of many aspects of the outbreak and response. However, it lacks the focus that would make it a useful scientific paper. For example, it is noted that the outbreak occurred after a ski event at the local ski resort. However, there is no real assessment of how that event contributed to spread across the county. The rural nature of the county is emphasized, as is the resort nature of the community. These raise separate, important issues with regard to transmission risk and control, and neither is fully addressed. 

Authors’ response: We have made several revisions based on reviewer #2’s feedback, including refining the focus and adding additional context and discussion regarding events and the impact of COVID-19 on rural communities. As our investigation was limited to cases reported South Central Public Health District in Idaho, we were not able to assess how exposures at the ski event might have contributed to national spread of COVID-19. However, we have noted this limitation in the discussion (lines 322-325). We have added some examples from the literature on how events (including travel to ski resorts and beer festivals in Germany and a wedding in rural Maine) and second homes contributed to COVID-19 spread in rural areas (lines 252-270).

The abstract describes the occurrence of 452 cases during March 13-April 10. However, the epicurve (figure 1) identifies more than 60-70 cases with onset of symptoms before March 13. The epi-curve and the descriptions of the outbreak are difficult to reconcile.

Authors’ response: March 13 to April 10th refer to the time frame during which COVID-19 cases were reported to South Central Public Health District, whereas the epi curve displays cases by date of symptom onset. In our study population, the median lag time between the date of symptom onset and the report date for a positive SARS-CoV-2 test was 13 days. This result has been added to lines 169-171. We have edited the abstract to indicate that March 13-April 10 refers to report dates, whereas illness onset began March 1. 

 I think that it would be useful to focus on the impact of the ski resort in the initiation of the outbreak. It would also be useful to highlight the interactions of the control measures on the epi-curve (fig. 1). If the epi-curve could be labeled with all of the key time line events, (ski festival, school closure, resort closure, county isolation order, state isolation order, travel restriction) it would provide a clear visual guide to the progression of the outbreak. This would be a better way to relate control measures to case incidence than is achieved in figure 3. The use of cumulative frequency data make it difficult to track the rate of new case onsets. It would be an even bigger bonus if the cases identified by sequence cluster could also be indicated on the epi-curve. I recognize that may not be feasible.

Authors’ response: As suggested, we have added the key time line events to the epi-curve (Figure 1). However, we did not include sequence information on this Figure as it would make the figure overly complicated and difficult to interpret. We have moved the figure showing cumulative case counts (previously Figure 3) to the supplementary information.

It seems that approximately 1/3 of the population of the county at any given time may be made up by visitors from out of state. It would be useful to try to estimate the rate of illness in this ephemeral group, or at least try to better estimate the impact of visitors on transmission dynamics. It is critical for similar resort communities to try to plan for potential disease introductions from tourists, and to have plans for surveillance and control of spread related to these temporary residents. It may be true that such cases are not typically counted as cases in the temporary community, but that does not diminish the importance of being able to track and account for them.

Authors’ response: We agree that understanding the role of visitors in disease transmission is important in resort communities. Surveillance of visitors and temporary residents would be difficult to implement using existing reportable disease systems, which are based on primary residence, and we were not able to conduct these type of estimates retrospectively for Blaine County. We have added some additional text to the discussion highlighting issues relating to temporary residents and recommend improving strategies to detect and monitor SARS-CoV-2 infections among temporary residents (lines 267-270). 

There are descriptions of definitions of close contacts on p.4, and changes in the definition. There does not seem to be any use of this data in the paper. If it is not being used for analytical purposes, is it needed in the methods? 

Authors’ response: Although we did not use contact tracing data for analytical purposes, we do report results on the number of close contacts monitored, what proportion of close contacts developed COVID-19 or COVID-19-compatible symptoms, and hours spent by volunteers conducting contact monitoring. For these reasons, we think defining close contacts in the methods section will be helpful to readers.

Similarly, there is substantial description of census blocks in the methods. The results do not present more than a cursory analysis of the census block data, that could be omitted. How do the census blocks relate to housing for guests of the resort?

Authors’ response: We have moved the description of census block analysis to the supplementary information. The aim of conducting the census block analysis was to determine if cases among Blaine County residents were limited to a particular geographic region or certain socio-economic groups within the county; results indicated that this was not the case. We did not relate census blocks to guests of the resort or other temporary residents; this might be difficult to conduct as visitor accommodation includes multiple hotels as well as Air BnB and other informal rental properties located throughout the county.

The introductions and discussions are very broad. Given the vast literature that is being published on the pandemic, a narrower focus on the impact of the ski resort on the community would enhance the interest and usefulness of this paper.

Authors’ response: We have modified the introduction and discussion and removed some of the more broad content and references. We agree that the impact of the ski resort is an important component of this paper. We also want to highlight two other key findings relevant to rural settings, namely that the high number of affected health care workers led to the temporary, partial closure of the only hospital in the county, and the long turnaround times for COVID-19 test results. Together, these findings demonstrate challenges faced by rural areas during the early stages of the COVID-19 pandemic.

---

## [Decision Letter · Decision Letter 1]

6 Apr 2021

Investigation and public health response to a COVID-19 outbreak in a rural resort community — Blaine County, Idaho, 2020

PONE-D-21-03832R1

Dear Dr. Dunne,

We’re pleased to inform you that your manuscript has been judged scientifically suitable for publication and will be formally accepted for publication once it meets all outstanding technical requirements.

Kind regards,

Shinya Tsuzuki, MD, MSc

Academic Editor

PLOS ONE

Reviewers' comments:

Reviewer's Responses to Questions

**Comments to the Author**

1. If the authors have adequately addressed your comments raised in a previous round of review and you feel that this manuscript is now acceptable for publication, you may indicate that here to bypass the “Comments to the Author” section, enter your conflict of interest statement in the “Confidential to Editor” section, and submit your "Accept" recommendation.

Reviewer #1: All comments have been addressed

Reviewer #2: All comments have been addressed

2. Is the manuscript technically sound, and do the data support the conclusions?

Reviewer #1: Yes

Reviewer #2: Yes

3. Has the statistical analysis been performed appropriately and rigorously? 

Reviewer #1: Yes

Reviewer #2: Yes

4. Have the authors made all data underlying the findings in their manuscript fully available?

Reviewer #1: Yes

Reviewer #2: Yes

5. Is the manuscript presented in an intelligible fashion and written in standard English?

Reviewer #1: Yes

Reviewer #2: Yes

6. Review Comments to the Author

Reviewer #1: Thank you for addressing my comments and the other reviewer's comments. This paper is much stronger now.

Reviewer #2: The authors have addressed reviewer's comments. The figure with the control measures imposed on the epi-curve is very nice.

7. PLOS authors have the option to publish the peer review history of their article (what does this mean?). If published, this will include your full peer review and any attached files.

Reviewer #1: No

Reviewer #2: No

---

## [Editor Report · Acceptance letter]

12 Apr 2021

PONE-D-21-03832R1 

Investigation and public health response to a COVID-19 outbreak in a rural resort community — Blaine County, Idaho, 2020 

Dear Dr. Dunne:

I'm pleased to inform you that your manuscript has been deemed suitable for publication in PLOS ONE. Congratulations! Your manuscript is now with our production department. 

Kind regards, 

on behalf of

Dr. Shinya Tsuzuki 

Academic Editor

PLOS ONE